# A Comparison of Inertial Measurement Unit and Motion Capture Measurements of Tibiofemoral Kinematics during Simulated Pivot Landings

**DOI:** 10.3390/s22124433

**Published:** 2022-06-11

**Authors:** So Young Baek, Mirel Ajdaroski, Payam Mirshams Shahshahani, Mélanie L. Beaulieu, Amanda O. Esquivel, James A. Ashton-Miller

**Affiliations:** 1Department of Mechanical Engineering, University of Michigan, Ann Arbor, MI 48109, USA; mirshams@umich.edu (P.M.S.); jaam@umich.edu (J.A.A.-M.); 2Department of Mechanical Engineering, University of Michigan-Dearborn, Dearborn, MI 48128, USA; majdaros@umich.edu (M.A.); aoe@umich.edu (A.O.E.); 3Department of Orthopedic Surgery, University of Michigan, Ann Arbor, MI 48109, USA; mbeaulie@umich.edu

**Keywords:** ACL injury, inertial measurement units (IMU), motion capture, jump landing, knee kinematics

## Abstract

Injuries are often associated with rapid body segment movements. We compared Certus motion capture and APDM inertial measurement unit (IMU) measurements of tibiofemoral angle and angular velocity changes during simulated pivot landings (i.e., ~70 ms peak) of nine cadaver knees dissected free of skin, subcutaneous fat, and muscle. Data from a total of 852 trials were compared using the Bland–Altman limits of agreement (LoAs): the Certus system was considered the gold standard measure for the angle change measurements, whereas the IMU was considered the gold standard for angular velocity changes. The results show that, although the mean peak IMU knee joint angle changes were slightly underestimated (2.1° for flexion, 0.2° for internal rotation, and 3.0° for valgus), the LoAs were large, ranging from 35.9% to 49.8%. In the case of the angular velocity changes, Certus had acceptable accuracy in the sagittal plane, with LoAs of ±54.9°/s and ±32.5°/s for the tibia and femur. For these rapid motions, we conclude that, even in the absence of soft tissues, the IMUs could not reliably measure these peak 3D knee angle changes; Certus measurements of peak tibiofemoral angular velocity changes depended on both the magnitude of the velocity and the plane of measurement.

## 1. Introduction

There are approximately 200,000 anterior cruciate ligament (ACL) ruptures in the United States every year [1]. Impulsive 3D knee loads on the order of three to four times the body weight (BW) have been shown to cause ACL fatigue failure *in vitro* during repeated simulated jump landings; these combined a knee flexion moment with an internal tibial rotation moment and axial compression due to gravitational, inertial, and trans-knee muscle forces [2,3]. This type of 3D loading causes both transitory internal rotation of the tibia relative to the femur as well as a forward translation of the tibial plateau relative to the femoral condyle due to the patellofemoral mechanism, both of which are known to significantly increase ACL strain during a landing or cut [2,4,5,6]. If one defines these as “high ACL strain” (‘HAS’) loading cycles, one might be able to track the number of these HAS cycles imposed weekly by an athlete during practice and thereby limit the number and severity of HAS loading cycles before damage accumulates to the point of ACL failure due to material fatigue [7].

Traditionally, laboratory-based optoelectronic motion capture systems have employed skin markers to measure the relative 3D motions between the shank and the thigh during the athletic maneuvers associated with ACL injuries. Inertial measurement units (IMU), on the other hand, are not constrained to a laboratory setting and can also be worn simultaneously by multiple individuals. Each IMU consists of a tri-axial linear accelerometer, tri-axial rate gyroscopes that measure angular velocity, and a triaxial magnetometer to measure the orientation of the gyro [8]. The IMU is usually attached directly to the skin or via a Velcro or elastic strap or garment around the body segment [9,10]. IMUs have been incorporated into many wearable sensor systems in order to track joint kinematics during gait analysis, sports, and rehabilitation activities [9,11,12] because they have the advantage of being small, light, and easily attachable to different body segments. It is possible that they could also be used to identify and count HAS loading cycles, but they would first have to be demonstrated to have acceptable accuracy and reproducibility in the field during impulsive loading maneuvers.

In terms of accuracy and repeatability, motion capture systems and IMUs have been used to validate one another’s results [11,13]. In particular, a motion capture system lends itself to measuring joint angles because it collects location data of markers via line of sight [14]. Angular velocities and accelerations calculated from these data are derived by mathematical differentiation, which is a process that introduces noise on the signal. On the other hand, because an IMU contains three orthogonal rate gyroscopes, it has the potential to directly measure angular velocities, and only a single differentiation or integration step is required to obtain acceleration and position data, respectively. Therefore, motion capture systems and IMUs each have their strengths and weaknesses.

During dynamic tasks such as a jump landing, it is well known that the skin, as well as the underlying subcutaneous fat and muscle, will vibrate after ground contact due to their mass and viscoelastic coupling to the underlying bone. When the IMU or marker is fastened directly to the skin, this vibration leads to a soft tissue motion artifact that degrades the kinematic recordings of what are meant to represent motions of the underlying bone [15,16]. Several authors have employed active or passive skin markers tracked by motion capture systems to evaluate the accuracy of measuring joint angles or joint angular velocities [10,13,17]. There have also been studies with magnetic resonance imaging (MRI) [18] and X-ray fluoroscopy measurements [15] of the kinematics of the underlying bone to remove the effect from soft tissue, but stereoradiography requires a static setup of the X-ray sources in the laboratory, is invasive and unethical for use on children or adolescents, and is completely impractical for use in the field.

There are published comparisons of quasistatic IMU outputs with known changes in angles measured, for example, by a coordinate measuring system [19]. However, we are not aware of studies that have compared IMU measurements of 3D changes in the knee angle or body segment angular velocity with those made with a motion capture system during a dynamic activity such as a jump landing. Moreover, none appear to have been made in the absence of a soft tissue artifact. However, such measurements can provide a baseline for knowing the conditions under which the IMU and motion capture systems can provide reliable data. If one removes the soft tissue, for example, cadaver experiments provide just such an opportunity to compare the IMU measurement of 3D knee angle changes with those measured using a standard motion capture system. Likewise, one can make a similar comparison of 3D tibial and femoral angular velocity changes measured via a pair of IMUs (which we, in this paper, consider the gold standard because of measuring angular velocity directly) with those measured by differentiating the motion analysis system angular data. Since the IMUs and marker triads can be mounted directly on the cadaveric tibia and femur with no intervening soft tissue, this setup presents the opportunity to compare the best possible 3D performance of the two measurement systems in the absence of a soft tissue artifact.

The goal of this paper was to test the null hypothesis that motion capture system and IMU measures of abrupt 3D changes in the tibial–femoral angle and angular velocity during a simulated unipedal jump landing are comparable. We tested that hypothesis using a Bland–Altman analysis to calculate the “limits of agreement” (LoAs) between the two measurement system results: the gold standard being the motion capture system displacement data and IMU angular velocity data.

## 2. Materials and Methods

### 2.1. Specimen Procurement and Preparation

A total of nine knees were harvested from six male and three female human donors (Table 1) procured from the University of Michigan Anatomical Donation Program, as well as Anatomy Gifts Registry, Science Care, and Gift of Life Michigan. The knee specimens were double-bagged and stored in a freezer at −20 ℃, until being removed from the freezer 48 h prior to dissection and testing. Thawed knees were dissected at room temperature, leaving intact the ligamentous capsular structures and the tendons of the quadriceps, medial hamstrings, lateral hamstrings, medial gastrocnemius, and the lateral gastrocnemius muscles. Following dissection, the distal tibia/fibula and proximal femur were cut to a standard 20 cm length from the center of the knee joint (palpated on the medial meniscus). The proximal femur and distal tibia were then potted in polymethylmethacrylate cylinders ready for mounting in the testing apparatus.

### 2.2. Testing Apparatus

Each cadaveric knee was mechanically tested using the Withrow-Oh [3,20] testing apparatus (Figure 1) that simulated repeated single-leg pivot landings. In each trial, in order to achieve a high enough loading rate to properly simulate a jump landing, a known weight (W, Figure 1) was released to impact the distal end of the tibia through a viscoelastic bumper whose properties were tuned to shape the temporal history of the impact force to peak at 70 ms [20]. Five preconditioning trials were used to find the drop height and weight required to simulate a three to four times body weight peak impulsive force on that knee. The loads applied to the knee were measured using a six-axis (MC3A-1000, AMTI, Watertown, MA, USA) load cell (L, Figure 1) located in series with the distal femur, whereas the reaction loads were measured simultaneously via an identical load cell located in series with the proximal tibia: these cells monitored the 3D input forces and moments, and 3D reaction forces and moments, respectively. The knee was initially set up with 15° of flexion held there by the quadriceps pretension, which reflects the pretension in the muscle prior to landing a jump. Each specimen was subjected to a maximum of 100 loading trials or as many trials as could be completed until the ACL failed. Failure was defined as a 3 mm or more increase in cumulative anterior tibial translation. The design of the apparatus caused the dropped weight to apply, simultaneously, an impulsive compression load, a knee flexion moment, and an internal tibial torque (T, Figure 1) to the knee, combined with realistic trans-knee muscle forces (Q, G, and H, Figure 1). Clamps were attached to grasp each tendon, with only the quadriceps tendon clamp requiring cryocooling to prevent slippage; active muscle tensile elasticity was simulated by a length of 2 mm diameter woven nylon rope, and each muscle construct was pre-tensed as follows: quadriceps (Q: 180 N), hamstrings (H: 70 N each), and gastrocnemius (G: 70 N each) [21,22].

An optoelectronic imaging system (Optotrak Certus, Northern Digital Inc., Waterloo, ON, Canada) was used to record the 3D tibiofemoral marker triad kinematics at 400 Hz with an accuracy of 0.2 mm RMS [23]. A set of three markers (‘Markers’, Figure 1), affixed to a rigid body, was attached to the distal femur and proximal tibia, respectively. A 3D digitizer was used to identify standard anatomical landmarks on the knee joint (‘Virtual markers’, Figure 1) to calculate relative and absolute 3D tibiofemoral kinematics throughout each trial. The motion capture and load cell data were automatically synchronized by the Optotrak system.

A wearable IMU (OPAL, APDM Inc., Portland, OR, USA, Figure 1) was securely attached adjacent to the triad near the distal end of the medial tibia and the proximal end of the mid femur using tensioned Velcro straps. Linear acceleration and angular velocities were measured at 128 Hz using the two IMUs. After being set up, the knee was first fully extended for the initialization process of the two APDM OPAL units to set initial conditions and minimize the influence of drift during the angular calculations using the two IMUs. The two IMUs then provided continuously recorded data on the triaxial linear accelerations and triaxial angular velocities of each bone until the experiment ended. 

Both the Optotrak system and APDM sensors have been validated in the literature for given applications [24,25,26].

### 2.3. Data Analysis

A total of 852 simulated pivot jump landing trials were conducted across all nine specimens. The 3D tibial and femoral bone motions were tracked using both the Certus motion capture systems and two APDM IMUs.

#### 2.3.1. Knee Joint Angle Calculations

The 3D tibiofemoral triad marker position data from the Certus system were filtered using a 4th order Butterworth low-pass filter with 20 Hz cutoff frequency implemented using MATLAB (MATLAB_R2019b; MathWorks, Natick, MA, USA). A custom MATLAB code [20] was used to calculate relative angles and translation using the method developed by Grood and Suntay [27].

The IMU data from each trial were used to calculate the knee joint angles using the proprietary sensor fusion algorithm provided by APDM Opal to determine the orientation of the devices using the method of Watanabe et al. [8,28].

#### 2.3.2. Comparison of the Knee Angles between the Two Measurement Methods

The two measurement systems acquire data at different sampling rates, so linear interpolation was used to match the IMU sampling rate to that of the motion capture system. The peak angle changes from the optoelectronic motion capture system were calculated as the difference between the initial and the maximum angle measured (Figure 2a). Similarly, the peak IMU angle changes were obtained from the difference between initial resting state and the maximum values (Figure 2b) to obtain the peak angles during each pivot landing.

The difference in peak knee angle change measured with the IMUs and the motion capture systems was calculated in each plane. Bland–Altman plots were then used to assess the variation in the IMU-derived peak knee angle change compared to the peak knee angle change measured using the Certus motion capture system. The mean of the difference (bias) and the 95% confidence interval (CI) of the bias (limits of agreement) were also calculated.

#### 2.3.3. Peak Tibial and Femoral Angular Velocity Change Calculations

The three orthogonal rate gyroscopes in the wearable IMUs on the tibia and femur directly measured the time histories of the angular velocities during each impulsive loading cycle. For the Certus, the 3D angular velocities in each orthogonal plane were calculated from the 3D motions of the femur and tibia Certus marker triads sampled at 400 Hz, as described in the the Section 2.3.1. The three markers on each bone were oriented in the motion capture system frontal plane, to establish a sagittal plane for the bones. These three markers on each bone allowed for the generation of unit directional vectors in each axis: anterior–posterior, medial–lateral, and proximal–distal. The temporal changes in angular rotations were then calculated in each direction using a five-point differential method and a fourth order low-pass Butterworth filter and 25 Hz calculated cut off frequency [29].

#### 2.3.4. Comparison of the Peak Tibial and Femoral Angular Velocity Changes Using the Two Measurement Systems

Using the measured peak angular velocity changes from the wearable IMUs, we compared the measured peak bone angular velocity changes from the Certus system calculated by differentiating the angular position data over the same time span of the simulated jump landings. To compare the peak values of the bone angular velocity changes, the averaged means of all testing trials for each system and standard deviation were calculated, as well as the percent error between the peak angular velocity changes calculated from the Certus system and those measured from the IMUs. Bland–Altman plots were used to view the limits of agreement between the bone angular velocities calculated from the Certus system and compared to the IMU rate gyro-based measurements.

## 3. Results

### 3.1. Measured Peak Knee Angle Changes

Bland–Altman plots comparing IMUs versus the motion capture system for measuring the knee flexion, knee internal rotation, and knee valgus angles are shown in Figure 3. The measured peak knee angle changes from the IMUs displayed systematic bias in that they were underestimated in two of the three planes. As the peak angle change increased, the bias decreased, with internal rotation exhibiting the largest peak angle change, but only −0.18° of bias. However, the width of the limits of agreements (LoAs) band increased, with large peak angle changes (Table 2). For all three orthogonal knee angles, the standard deviation (SD) of the angular differences (IMUs-Certus) as a percentage was ~20% in all three directions; specifically, 19.8%, 25.4%, and 18.3%, respectively.

### 3.2. Comparison of the Tibial and Femoral Angular Velocity Changes Calculated from Certus System Data with IMU Data

The Bland–Altman plots comparing the motion capture system versus IMUs outcomes for angular velocities by plane for each bone are shown in Figure 4. The bias calculated for the femur in all directions was less than that measured for the tibia: 72.7°/s (tibia) vs. −35.8°/s (femur) in the sagittal plane, −159.1°/s (tibia) vs. 112.0°/s (femur) in the transverse plane, and 536.3°/s (tibia) vs. 100.4°/s (femur) in the frontal plane (Figure 4 and Table 3). The width of the LoA band was considerably less in the sagittal plane (54.9°/s for the tibia and −32.5°/s for the femur) than in the other planes.

The bone angular velocity changes in the transverse plane exhibited the largest magnitude (Table 3). The average of the IMU-measured angular velocity changes was −832.5°/s for the tibia, and the difference in the averages measured using the IMUs and those calculated from the Certus system exhibited only an 18.9% difference, with an SD of 7.3%.

## 4. Discussion

There are two novel aspects to this study. First, we made kinematic measurements over a time course that is characteristic of a limb landing on the ground during which time injury can result. The first peak in the foot–ground reaction force usually occurs 35–50 ms after ground contact [30,31]. However, the musculoskeletal response of the more proximal parts of the lower extremity limb, of course, lags that peak. For example, if one considers the mechanism of ACL injury [7], the magnitude of the increase in ACL strain is proportional to the change in the knee flexion angle (via the patellofemoral mechanism) and the change in the internal tibial rotation angle, which both peak ~70 ms following ground contact [2,20]. That is why we focused on calculating the change in angle and change in angular velocity over that time interval in this study. Second, we could find no studies comparing IMU and motion capture system performance measuring the kinematics of a human body segment, knee joint peak angle change, or peak bone angular velocity changes in the absence of a soft tissue motion artifact. The significance is that our experiment provides the opportunity to assess the best possible dynamic performance of the IMU and motion capture systems without the artifact caused by soft tissue vibration. Any in vivo measurements are certain to exhibit more variability because of noise generated by soft tissue movement artifacts.

Bland and Altman originally developed their eponymous method to compare an experimental measurement method with a standard measurement method by introducing the limits of agreement approach [32]. It has since been adopted for use by many fields [33]. The method is useful because, unlike a correlation coefficient, it quantifies any bias present, as well as the limits of agreement, within 95% of the differences for the experimental method compared to the standard method. Of course, the narrower the limits of agreement, the better the experimental method is, until the results become interchangeable with one another [34]. The peak angle changes in each orthogonal plane were slightly underestimated by the IMU measurements. Even though the bias was only −0.6% for internal rotation, which exhibited the largest peak angle change of ~16°, the peak knee angle difference between measurement methods amounted to ~±40% of the LoA (calculated from 1.96×-SD in all three orthogonal planes). This result indicates that the knee angle change calculated using these APDM IMUs and their fusion algorithm could not reliably capture the peak kinematics of the weight acceptance phase of a movement as dynamic as a unipedal pivot landing. This was true in the absence of a soft tissue artifact, so the presence of such an artifact in vivo would only make matters worse. 

The orientation of the Certus camera relative to the lower extremity was found to affect the calculated angular velocity changes. This was because the marker triads were essentially placed in the knee parasagittal plane parallel with the frontal plane containing the three optoelectronic cameras that constitute the Certus system. This provided its best resolution, but the resolution is lower in the Certus cameras’ depth plane, orthogonal to its frontal plane [23]. The Certus camera orientation therefore explains the narrow LoA band in the sagittal plane for the Certus-measured angular velocities. With the present setup, when the tibia was forced to internally rotate relative to the femur during the pivot landing, the tibial marker triad moved in the Certus camera depth plane, whereas the femoral triad moved principally in the Certus frontal plane, hence the smaller bias we found in the results for the femur than for the tibia. The exception was internal tibial rotation in the transverse plane, perhaps because it had a larger angular velocity change, 832.5 °/s, compared to the motion in the other planes. The upshot is that the orientation of the Certus camera relative to the subject will affect the measurement accuracy of certain knee angles. However, as can be seen from the results for internal rotation, if angular velocities are sufficiently high, the camera orientation was less of an issue.

A first limitation was that the sampling frequencies of the two measurement systems were different. This could have affected the accuracy of the results because of the need for interpolation between data points when resampling data at the same rate as the second measurement system. However, the changes in angle and angular velocity were relatively smooth, so this should not have affected the results materially. A second limitation was that there will have been a slight phase difference between when the impact force acts distally on a limb and when it acts more proximally on the knee [35]. In this experiment, the order of sensed displacements would have been the tibia marker triad first (because of being closer to the point of impact), then the tibial IMU, then the femoral IMU, then the femoral marker triad. Such phase differences likely had a minor effect on our results because the angle and angular velocities of each system were calculated relative to the peak values for each trial. Finally, we employed the generic APDM software algorithm to process these data, but a custom-tuned algorithm can improve the results slightly, particularly by reducing the phase lag in the generic APDM results due to the rapid acceleration profiles [36] associated with HAS cycles. In summary, we do not believe that these limitations materially affected the overall results in comparing the two measurement systems.

## 5. Conclusions

Due to the fact that the LoAs ranged from 35.9% to 49.8% of the measured joint angle, the APDM IMUs could not reliably measure the sudden changes in joint angles that occur during simulated pivot landings, even in the absence of soft tissue artifact. Therefore, improvements to these IMUs are needed to be able to reliably measure HAS cycles *in vitro*, let alone in the presence of soft tissue in vivo.

## Figures and Tables

**Figure 1 sensors-22-04433-f001:**
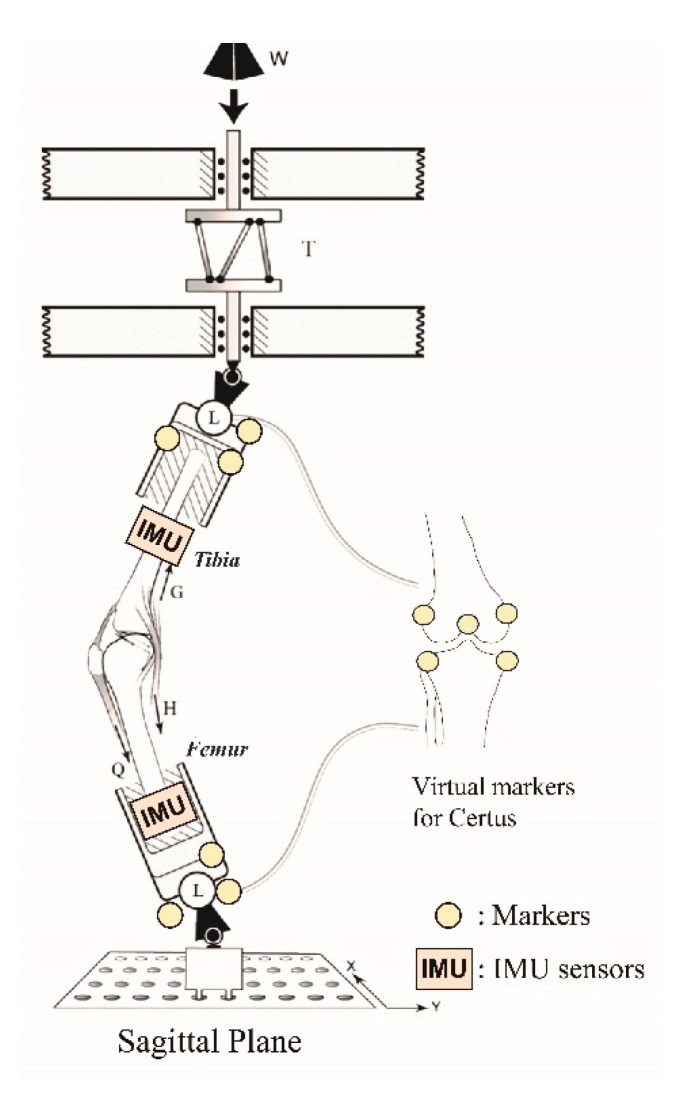
Diagram of the Withrow-Oh testing apparatus [3,20] showing the locations of the fixed active marker triads along with the locations of the virtual active markers obtained by manual digitization prior to the experiment. Modified, and reproduced with permission from [3]. 2011, The Journal of Bone and Joint Surgery, Inc.

**Figure 2 sensors-22-04433-f002:**
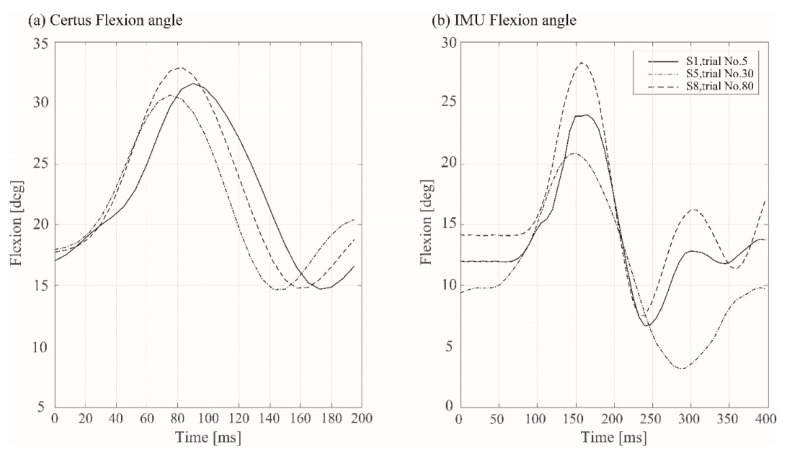
Three examples of the time course of the changes in knee flexion angle during simulated jump landings. Panel (**a**) shows measurements from the optoelectronic motion capture system, whereas panel (**b**) shows the calculated angles from the accelerometer and gyroscopic data from the IMUs. The peak angle change from the IMUs was calculated as the difference between the initial resting state and the maximum angle measured.

**Figure 3 sensors-22-04433-f003:**
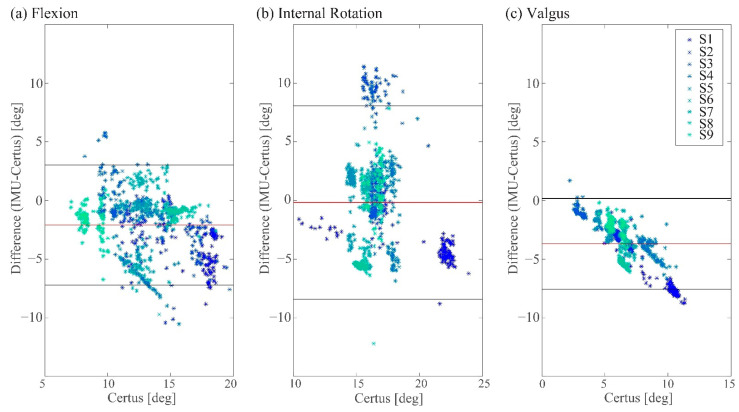
Bland–Altman plots comparing the measurements from the two IMUs with those of the motion capture system for measuring peak knee angle change in each direction during simulated jump landings. In this and the following figure, the different colors represent data from different knee specimens (S).

**Figure 4 sensors-22-04433-f004:**
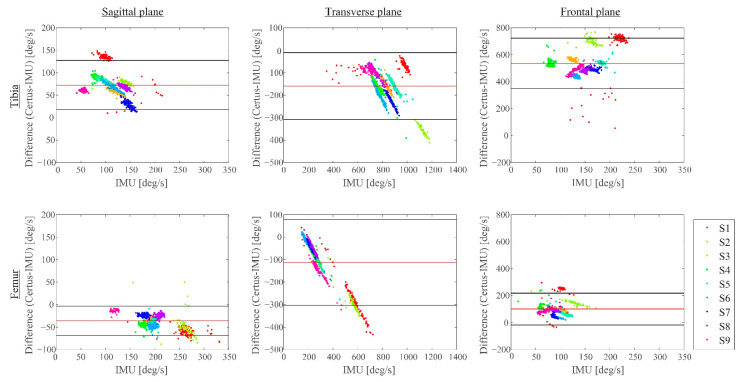
Bland–Altman plots showing differences, by plane, between Certus system and IMU in the angular velocity change for the tibia (top row) and the femur (bottom row). The x-axis shows the angular velocities measured via the IMUs, whereas the y-axis shows the difference in angles for each plane (left, center, and right panels). The bias was negative in the transverse plane, positive in the frontal plane, and mixed in the sagittal plane for the two bones.

**Table 1 sensors-22-04433-t001:** Demographic data of the knee specimen donors.

Specimen No.	Gender	Side	Age [years]	Weight [kg]	Testing Condition
S1	F	L	20	86.6	3 BW 100 trials
S2	F	R	28	63.5	4 BW 100 trials
S3	F	R	30	82.1	3 BW 52 trials *
S4	M	L	39	54.4	4 BW 100 trials
S5	M	R	32	68.0	4 BW 100 trials
S6	M	R	32	88.5	3 BW 100 trials
S7	M	R	25	86.2	3 BW 100 trials
S8	M	R	31	92.5	3 BW 100 trials
S9	M	R	33	49.9	4 BW 100 trials
Total					852 trials

F: Female, M: Male, L: Left, R: Right, BW: Body Weight. * Testing was halted due to ACL failure, defined as a 3 mm increase in cumulative anterior tibial translation.

**Table 2 sensors-22-04433-t002:** Mean (SD) peak angle changes measured via Certus and IMUs, along with the corresponding Bland–Altman bias and limit of agreement results calculated according to the formula in the table footnote at bottom right.

	Flexion	Internal Rotation	Valgus
Certus	13.3 (2.9)°	16.7 (2.1)°	6.6 (2.1)°
IMU	11.2 (3.2)°	16.6 (4.3)°	2.9 (1.1)°
Bias	−2.1°	−0.2°	−3.0°
LoA	(−7.2°, 3.0°)	(−8.4°, 8.1°)	(−7.6°, 0.2°)
Diff [%]	−15.3% (19.8%)	−0.6% (25.4%)	−52.8% (18.3%)
Agreement Range [%]	±38.8%	±49.8%	±35.9%

Bias = $\frac{\sum^{\;}\left(IMU-Certus\right)}{N}$, LoA = Bias±1.96×SD, Diff [%] = $\sum^{\;}\left(\frac{IMU-Certus}{Certus}\right)\times 100/N$, Agreement Range [%] = 1.96×SD of Diff [%].

**Table 3 sensors-22-04433-t003:** Mean (SD) values for Bland–Altman comparisons of peak angular velocity from Certus and IMUs for the three planes. On average, the Certus measurements tended to overestimate the IMU measurements. The percentage difference (Diff [%]) is defined in the table footnote.

	Certus [°/s]	IMU [°/s]	Certus-IMU [°/s]	LoA [°/s]	Diff [%]
Sagittal plane					
Tibia	182.3 (32.6)	109.7 (31.5)	72.7 (28.0)	(17.8, 127.5)	75.4% (38.6%)
Femur	159.7 (33.0)	195.5 (41.7)	−35.8 (16.6)	(−68.3, −3.3)	−17.9% (6.5%)
Transverse plane					
Tibia	673.4 (103.2)	832.5 (120.2)	−159.1 (75.9)	(−308.0, −10.3)	−18.9% (7.3%)
Femur	188.8 (45.3)	300.8 (131.6)	−112.0 (97.1)	(−302.3, 78.2)	−31.8% (17.4%)
Frontal plane					
Tibia	684.9 (117.5)	148.6 (38.4)	536.3 (95.8)	(348.5, 724.1)	384.4% (121.6%)
Femur	191.3 (65.1)	90.9 (19.2)	100.4 (60.2)	(−17.6, 218.4)	115.5% (75.3%)

LoA denotes limits of agreement; Diff [%] = $\frac{\mathrm{Certus}–\mathrm{IMU}}{\mathrm{IMU}}$.

## Data Availability

Not applicable.

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
