# Peer review of "A Comparison of Inertial Measurement Unit and Motion Capture Measurements of Tibiofemoral Kinematics during Simulated Pivot Landings"

_sensors, 2022, doi:10.3390/s22124433_

Round 1
Reviewer 1 Report
Thank you for the opportunity to review this manuscript.
The study is interesting, and the results were clear. However, the authors should re-consider and add some explanations in the Introduction and the Methods section. The authors repeated “the gold standard” quite a few times to explain for the measurement, but the authors can explain it once in the Introduction and other repetitions should be omitted. Please add and/or provide the reliability and the validity of the sensors. The statistical results of Bland-Altman plots should be explained (the significance) clearly in the text or the Figures.
Line 18-19: The sentence is unclear. Please rewrite.
Line 48: …a landing or “cutting”
Line 66: It is possible “that”…
Line 73: “noisy” is an awkward word so please change the word.
Table 1: Please add abbreviations below the table.
Line 152: Are there any references for setting up the amount of force applied? Or any back support data for this?
Line 165: How did both measurements synchronize? Please add more explanation.
Line 240: Missing “Figure 2.” The Figure 3 should be Figure 2.
Line 250-251: Are the formula with Table 2? If so, the font size should be modified.
Line 262, 266; The Figure 4 should be Figure 3.
Table 3: Please add abbreviation below the table.
Line 302: The sentence should be rewritten (grammatically incorrect).
Author Response
We thank the reviewers for their careful reviews and the points raised. Please find our responses below, and you can turn on tracked changes in the attached document to see changes within the context of the manuscript.
Point 1: The authors repeated “the gold standard” quite a few times to explain for the measurement, but the authors can explain it once in the Introduction and other repetitions should be omitted.
Response 1: Thank you for pointing this out, we agree. We have sharply reduced the times “gold standard” is used: it now only appears twice in the Abstract and twice in the main text.
Point 2: Please add and/or provide the reliability and validity of the sensors.
Response 2: We agree.
Concerning the reliability of the Optotrak system and APDM IMU, in Line 164 we have added “Both the Optotrak system and APDM sensors have been validated in the literature for given applications.” with the following citations.
Richards J. The measurement of human motion: A comparison of commercially available systems. Hum. Mov. Sci.,1999; 18 :589–602
Small CF, Bryant JT, Dwosh IL, Griffiths PM, Pichora DR, Zee B. Validation of a 3D optoelectronic motion analysis system for the wrist joint. Clin Biomech. 1996;11(8):481–483. doi:10.1016/S0268-0033(96)00042-3
Washabaugh EP, Kalyanaramanreliab T, Adamczyk PG, Claflin ES, Krishnan C. Validity and repeatability of inertial measurement units for measuring gait parameters. Gait Posture. 2017;55:87–93.
Point 3: The statistical results of Bland-Altman plots should be explained (the significance) clearly in the text or the Figures.
Response 3: The explanation is now in lines 225-231 and 247-253
Point 4: Line 18-19: The sentence is unclear. Please rewrite.
Response 4: Done. In line 21 we have deleted the word “corresponding” which was confusing as to its meaning.
Point 5: Line 48: …a landing or “cutting”
Response 5: This point has now become moot because this whole section was deleted in accordance with the reviewer’s wishes.
Point 6: Line 66: It is possible “that”…
Response 6: In line 55, we have now inserted “that” into the sentence.
Point 7: Line 73: “noisy” is an awkward word so please change the word.
Response 7: In line 62 we have altered this wording to clarify it.
Point 8: Table 1: Please add abbreviations below the table.
Response 8: Done.
We have added “F: Female, M: Male, L: Left, R: Right, BW: Bodyweight” below the table as requested
Point 9: Line 152: Are there any references for setting up the amount of force applied? Or any back support data for this?
Response 9: Yes. In line 142 we now cite two references to support the magnitude of the muscle forces applied:
Withrow TJ, Huston LJ, Wojtys EM, Ashton-Miller JA. The relationship between quadriceps muscle force, knee flexion, and anterior cruciate ligament strain in an in vitro simulated jump landing. Am J Sports Med. 2006;34:269-74.
Withrow TJ, Huston LJ, Wojtys EM, Ashton-Miller JA. Effect of varying hamstring tension on anterior cruciate ligament strain during in vitro impulsive knee flexion and compression loading. J Bone Joint Surg Am. 2008;90:815-23. 26.
Point 10: Line 165: How did both measurements synchronize? Please add more explanation.
Response 10: In line 154 we now state that the Optotrak system automatically synchronized the load cell and motion capture data.
Point 11: Line 240: Missing “Figure 2.” The Figure 3 should be Figure 2.
Response 11: We believe you are mistaken because Figure 2 was always present in the manuscript (Line 198 in the original version) and now is in Line 188 of the revised version.
Point 12: Line 250-251: Are the formula with Table 2? If so, the font size should be modified.
Response 12: Yes, the formulae are part of Table 2 so the font has now been modified so the fonts are consistent to make it obvious where they belong.
Point 13: Line 262, 266; The Figure 4 should be Figure 3.
Response 13: We believe you are mistaken because Figure 2 was always present (see Point 11 above) and so we have left the numbering of the figures as it was in the original manuscript.
Point 14: Table 3: Please add abbreviation below the table.
Response 14: We have now added “LoA: Limit of Agreement” below the Table.
Point 15: Line 302: The sentence should be rewritten (grammatically incorrect).
Response 15: We believe you are mistaken - the senior author is a native English speaker who grew up in England and the sentence is grammatically correct.

Reviewer 2 Report
Thank you for opportunity to review the paper entitled “A comparison of IMU and motion capture measurements of 2 tibiofemoral kinematics during simulated pivot landings”. The aim of the study was comparison of Certus motion capture and Inertial measurement unit measurements of tibiofemoral angle and angular velocity changes during simulated pivot landings of nine cadaver knees. Based on the research, the authors conclude that even in the absence of soft tissues, the IMUs could not reliably measure these peak 3D knee angle changes; Certus measurements of peak tibiofemoral angular velocity changes depended on both the magnitude of the velocity and the plane of measurement.
After reading the manuscript, I have to admit that the topic is important and the article written correctly. After a few minor corrections it will be suitable for publication in the journal.
The authors used the abbreviation (IMU) in the title of the article, which may be incomprehensible to many readers. This needs to be corrected.
In the abstract, the authors also use abbreviations without translating them. Keep in mind that most readers read the abstract first.
The introductory part is too long. It also contains a lot of text not entirely related to the topic of the work. The authors first focused on ACL injuries and the mechanism of its damage (a few paragraphs), and then the technical description of the measuring equipment (which should be included in the methodical part of the work). There is also no clearly formulated purpose for this work. The introductory part should be redrafted.
The methodological part of the work is its best part. The authors presented the entire experiment in detail. The results were also presented clearly.
In the discussion, the authors devote too much time to the description of the study, which is in the methodological part, and too little time to what was obtained from the study, and what significance it has in practice.
Requests should be redrafted. There are too many of them, they are too detailed. One overall conclusion from the research is missing.
Author Response
We thank the reviewers for their careful reviews and the points raised. Please find our responses below, and you can turn on tracked changes in the attached document to see changes within the context of the manuscript.
Point 1: The authors used the abbreviation (IMU) in the title of the article, which may be incomprehensible to many readers. This needs to be corrected.
Response 1: Thank you. We have deleted the abbreviation in the title and written it out as “inertial measurement unit”.
Point 2: In the abstract, the authors also use abbreviations without translating them. Keep in mind that most readers read the abstract first.
Response 2: At the first use of IMU in the Abstract, we now define this acronym.
Point 3: The introductory part is too long. It also contains a lot of text not entirely related to the topic of the work. The authors first focused on ACL injuries and the mechanism of its damage (a few paragraphs), and then the technical description of the measuring equipment (which should be included in the methodical part of the work). There is also no clearly formulated purpose for this work. The introductory part should be redrafted.
Response 3: Thank you for pointing this out. We agree and have considerably shortened the Introduction in response to your comments. Three and a half paragraphs have been removed along with other changes.
Point 4: In the discussion, the authors devote too much time to the description of the study, which is in the methodological part, and too little time to what was obtained from the study, and what significance it has in practice.
Response 4: We respectfully disagree. In order to understand what is new and significant about the study, we had to place the results we obtained within the context of both the experiment that was performed as well as the existing literature. In so doing we point out that current APDMs are unlikely to provide accurate measures of changes in knee angle or angular velocities during the part of the jump landing movement when the ACL ruptures in living people due to the fact they cannot reliably do it in cadaver knees (with no soft tissues interposed between the sensor and the bone). So they certainly cannot do it in the more challenging measurement situation when there are soft tissues between the sensors and the bone in living people. By implication, we are gently saying that the IMU sensor measurements that are being made today are unreliable in living athletes during these highly dynamic jump landing movements when injury occurs.
Point 5: Requests should be redrafted. There are too many of them, they are too detailed. One overall conclusion from the research is missing.
Response 5: We believe you are alluding to there being too many points in the original Conclusions section. So we now state one major conclusion.

Round 2
Reviewer 1 Report
The authors have thoroughly considered the suggestions and criticisms from the reviewers and revised the manuscript accordingly.
Reviewer 2 Report
Thank you for the opportunity to re-review the article entitled „A comparison of inertial measurement unit and motion capture 2 measurements of tibiofemoral kinematics during simulated 3 pivot landings”. After reading the article in the new, revised version and after reading the authors' responses to the review, it must be stated that the authors have significantly improved the manuscript. They also provided appropriate explanations. Therefore, I believe that the article in its current form qualifies for publication in the journal.